# Interaction between the assembly of the ribosomal subunits: Disruption of 40S ribosomal assembly causes accumulation of extra-ribosomal 60S ribosomal protein uL18/L5

**Nusrat Rahman[¤a], Md Shamsuzzaman[¤b], Lasse Lindahl[ID]***

Department of Biological Sciences, University of Maryland, Baltimore County (UMBC), Baltimore, Maryland, United States of America

¤a  Current address: Bentley Health Thought Leadership Network, Bentley University, Waltham, Massachusetts, United States of America
¤b  Current address: Verizon Innovation Center, Waltham, Massachusetts, United States of America
*  lindahl@umbc.edu

**Data Availability Statement:** All relevant data are within the manuscript and its Supporting Information files.

## Abstract

Inhibition of the synthesis of an essential ribosomal protein (r-protein) abrogates the assembly of its cognate subunit, while assembly of the other subunit continues. Ribosomal components that are not stably incorporated into ribosomal particles due to the disrupted assembly are rapidly degraded. The 60S protein uL18/L5 is an exception and this protein accumulates extra-ribosomally during inhibition of 60S assembly. Since the r-proteins in each ribosomal subunit are essential only for the formation of their cognate subunit, it would be predicted that accumulation of extra-ribosomal uL18/L5 is specific to restriction of 60S assembly and does not occur abolition of 40S assembly. Contrary to this prediction, we report here that repression of 40S r-protein genes does lead to accumulation of uL18/L5 outside of the ribosome. Furthermore, the effect varies depending on which 40S ribosomal protein is repressed. Our results also show extra-ribosomal uL18/L5 is formed during 60S assembly, not during degradation of mature cytoplasmic 60S subunits. Finally, we propose a model for the accumulation of extra-ribosomal uL18 in response to the abolition of 40S r-proteins.

## Introduction

The ribosome biogenesis process is preserved throughout eukaryotic evolution, although the complexity has evolved from yeast to humans [1, 2]. Most of the ribosome construction occurs in the nucle(o)lus where RNA polymerase I transcribes a long 18S-5.8S-25S/28S precursor rRNA (pre-rRNA) and RNA polymerase III transcribes 5S pre-rRNA [3–6]. Ribosomal proteins are translated in the cytoplasm and chaperoned into the nucle(o)lus, where they bind to the emerging ribosomal precursor particles concurrently with the processing of the pre-rRNA into the mature components. The ribosome manufacture is facilitated by more than 250

**Funding:** This study was funded by grant number 0920578 from the National Science Foundation, USA to JM Zengel and LL, and a gift from The Benelein Technologies, LLC to LL (no grant number). Further funding was provided by an internal appropriation from the University of Maryland, Baltimore County to LL (no grant number). The funders had no role in study design, data collection, and analysis, decision to publish, or preparation of the manuscript.

**Competing interests:** The authors have read the journal's policy and have the following competing interests: This study was funded in part via a gift from Benelein Technologies, LLC. This does not alter our adherence to PLOS ONE policies on sharing data and materials. There are no patents, products in development, or marketed products to declare.

ribosomal assembly factors, a number of which are important for the assembly of both the 40S and the 60S ribosomal subunits, while most are specific to the formation of one of the ribosomal subunits [7].

Whereas a subset of *Saccharomyces cerevisiae* (yeast) r-proteins is not required for the formation of ribosomes, most r-proteins are essential for assembly of their subunit [8]. A significant reduction of the production of just one essential r-protein or assembly factor prevents completion of the assembly of the cognate subunit, while the assembly of the other subunit continues (e.g. [9–11]). Moreover, abolishment of the assembly of a ribosomal subunit does not stop the synthesis of its r-proteins, but proteins that fail to become incorporated into stable ribosomal particles are rapidly eliminated by proteasomal turnover [10, 12, 13]. Nevertheless, one 60S protein, uL18, evades rapid degradation and accumulates in a complex with 5S rRNA outside of the ribosome when 60S assembly is abrogated by repression of uL5 synthesis [14]. [Note that uL5 was named L16 when this reference was published]. Since extra-ribosomal r-proteins are believed to play a role in regulation of factors controlling growth in metazoans [15, 16], it is important to understand the formation of extra-ribosomal r-protein pools.

Because the r-proteins in each ribosomal subunit are essential only for the assembly of their cognate subunit, it would be expected that interruption of the assembly of one subunit only affects the accumulation of extra-ribosomal r-proteins specific to that subunit. We tested this expectation by repressing several 40S r-protein genes and measuring the buildup of extra-ribosomal r-proteins. Surprisingly, and in contrast to the prediction, extra-ribosomal uL18 accrues when the synthesis of 40S r-proteins is constrained, but the amount of extra-ribosomal uL18 accumulating depends on which 40S r-protein gene is repressed. We interpret these results to mean that disruption of the assembly of the 40S subunit affects the kinetics, and perhaps even the pathway, of assembly of the 60S subunit. Furthermore, we show that protection of uL18 does not require the formation of the canonical 60S subunit assembly intermediate of uL18, uL5, 5S rRNA, and the Rrs1 and Rpf2 assembly factors.

## Materials and methods

### Nomenclature for r-proteins

The nomenclature of r-proteins has been changed twice since 1997 [17, 18]. We use the 2014 universal nomenclature. In the figures, the 1997 protein names are also indicated after a slash.

### Strains and growth conditions

All strains are derived from *S. cerevisiae* BY4741. In each strain one gene encoding r-proteins eS4, eS6, uS17, eS19, eS31, eL40, or eL43, or the 60S assembly factors Rrs1, or Rpf2 was expressed exclusively from the *GAL1/10* promoter (S1 Table). These strains are named $P_{gal}$-xx, where xx is the name of the protein expressed from the *gal* promoter. In the experiment shown in Fig 1B, $P_{gal}$-eL43 was transformed with a plasmid carrying a gene for uL18-FLAG expressed from the constitutive RpS28 promoter (Philipp Milkereit, personal communication).

Cells were grown at 30˚C with shaking in YEP-galactose medium. At $OD_{600}$ of 1.0–1.3 (about $2x10^7$ cells per ml), the culture was shifted to YPD (glucose) medium by diluting the galactose culture with 10 volumes of prewarmed glucose medium (starting $OD_{600}$ of 0.1–0.15). The glucose culture was then grown for 2–3 doubling times until a desired cell density ($OD_{600}$ 0.8–1) was reached. All strains have a doubling time of 1.5–2.0 hours in galactose, but the growth rate gradually decrease in a strain-specific manner after the shift to glucose medium due to the repression of r-protein genes [11]. The actual time each culture grew in glucose medium is indicated on the figures. Examples of growth curves and sucrose gradient $A^{260}$ profiles of crude extracts before and after the shift are shown in [11]. As shown previously a 55S

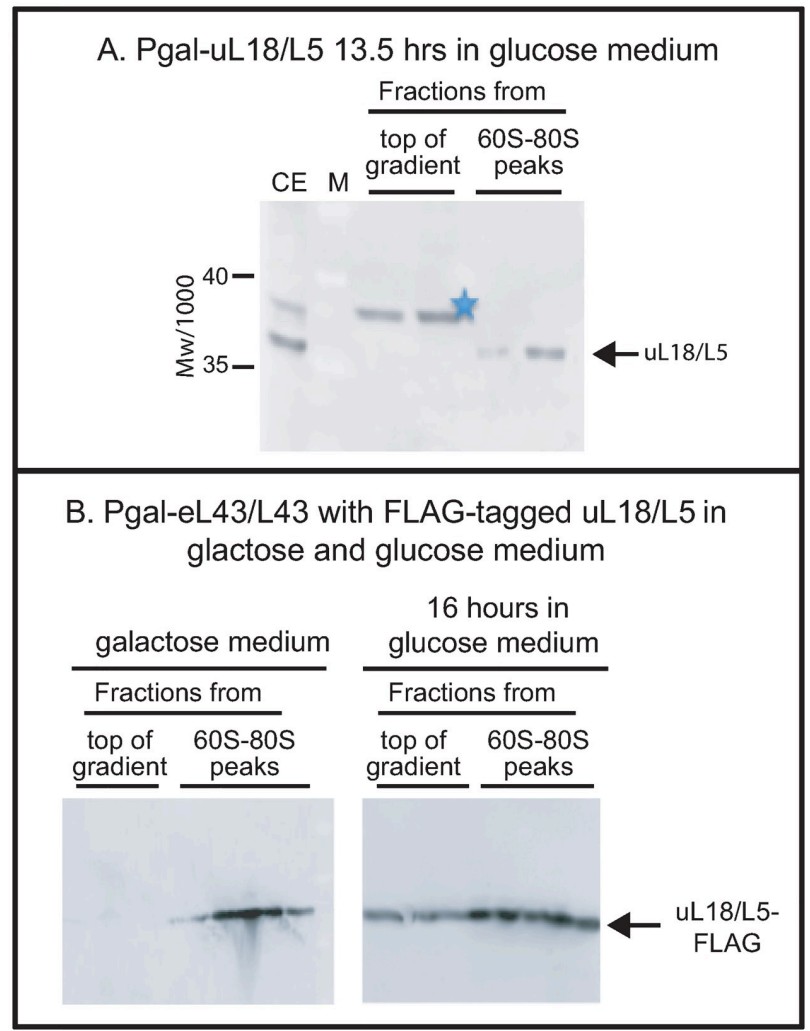

**Fig 1. Analysis of the specificity of anti-uL18/L5.** (A) The uL18/L5 reactive band seen close to the top of the sucrose gradient after repressing eL43/L43 or eL40/L40 formation (Figs 2 and 3) is absent after repressing uL18/L5 synthesis. $P_{gal}$-uL18/L5 was grown in galactose medium and shifted to glucose medium. A lysate prepared after repression of uL18/L5 gene for 13.5 hours was fractionated on a sucrose gradient and consecutive fractions from the top of the gradient and the 60S-80S ribosome peaks were analyzed by western blot stained with anti-uL18/L5. (B) Distribution of FLAG-tagged uL18/L5 (uL18/L5-FLAG) in sucrose gradients loaded with lysates prepared before and after repressing eL43/L43 synthesis. $P_{gal}$-eL43/L43 was transformed with a plasmid harboring a constitutively expressed gene for uL18/L5-FLAG. The resulting strain was grown in galactose medium and shifted to glucose medium for 16 hours. Lysates prepared from cells before and after the shift were fractionated on sucrose gradient and aliquots of consecutive fractions from the top of the gradient and the 60S-80S peaks were analyzed for content of FLAG-tagged protein by western blot. The western blots in this figure were not cropped. M: Molecular weight markers/1000. CE: Crude cell Extract.

ribosomal particle derived from the 60S accumulates forms beginning about 4 hours after the repression of 40S r-protein genes [11]. Here we have included the uL18 in the 55S peak as part of the "60S-80S ribosome fractions".

## Cell lysis and fractionation

Cells were harvested by centrifugation at 8000 rpm for 10 minutes and washed once with 10 mL ice-cold RNase free water and stored at -20˚C until further use. Procedures for lysis and

sucrose gradient centrifugation were described previously [11]. Each sucrose gradient was loaded with 20 A$^{260}$ units of crude cell extract cleared of debris. After centrifugation, the gradient was fractionated into 0.5 ml fractionations and 45 μl of the indicated fractions was used for western blot analysis.

### Western analysis and antisera

Western blots [11] were probed with rabbit polyclonal antisera prepared for our laboratory by Covance (Princeton, New Jersey, USA) using synthetic peptides with the sequence of 20–22 N-terminal amino acids of uS4, uL4, uL5, and uL18 as antigens. Each antiserum was titrated against increasing amounts of whole cell extract on western blots (see Figure S6 in ref [11] (https://www.life-science-alliance.org/content/2/2/e201800150#nogo). Based on the amount of lysate loaded on each sucrose gradient and the number of sucrose gradient fractions containing r-proteins, we determined that antisera are in excess over the r-proteins on our western blots. Monoclonal anti-FLAG antibody was purchased from Thermo-Fisher (catalog number MA1-91878). Blots were scanned in a Storm 860 Imager System (Molecular Dynamics) and the bands were quantified using ImageJ or Adobe Photoshop CC 2019.

As described in Results, western blots probed with anti-uL18 reacted with two proteins close to the top of sucrose gradients loaded with extracts of certain strains harvested after, but not before, a shift from galactose to glucose medium. One of these proteins co-migrated during electrophoresis with the ribosomal uL18 in the 60S-80S fractions, while the other had a slightly lower electrophoretic mobility. To determine if any of these proteins actually are related to uL18, we shifted P$_{gal}$-uL18 to glucose medium for 13.5 hours and fractionated a lysate on a sucrose gradient. Fractions from the top and the ribosome peaks of the gradient were analyzed on a western blot. As seen in Fig 1A, there was no uL18 reactive band comigrating with ribosomal uL18 after repressing the uL18 gene, even though the band was seen after repressing either eL40 or eL43 synthesis (see below). This confirms that the comigrating band at the top of the gradient seen after abolishing the synthesis of a number of r-proteins, other than uL18, actually is uL18. However, the slightly slower band (marked with star in all figures) was present even after repressing uL18, indicating that it may not be related to uL18. To test this, we transformed P$_{gal}$-eL43 with a plasmid constitutively expressing FLAG-tagged uL18 (uL18-FLAG) in addition to the native uL18 chromosomal gene. After shifting this strain to glucose for 16 hours, a single band of uL18-FLAG, co-migrating with the uL18-FLAG band in the ribosomal fractions, appeared at the top of the gradient (Fig 1B), but no band corresponding to the starred band in blots stained with anti-uL18 was seen. From these experiments, we conclude that the band marked with a star in the blots stained with anti-uL18 is not related to uL18, but rather an unspecific protein that cross-reacts with our uL18 antiserum. This is also supported by the presence of the starred band after shifting the parent strain to glucose medium (Fig 2C).

### Quantification of extra-ribosomal uL18

The fraction of the total uL18 found at the top of the sucrose gradient, i.e. the extra-ribosomal uL18, was determined by quantitative western blots. In some experiments, all gradient fractions were subjected to quantification, but in other experiments, we analyzed only two fractions from the top of the gel and three fractions from the 60S-80S region of the sucrose gradient. To compare the two methods of quantification, we used western blots that include all sucrose gradients fractions and compared the results of quantifying (i) uL18 at the top of the gradient/total uL18 across the gradient, and (ii) uL18 in two fractions at the top of the gradient/the sum of uL18 in two top fractions and uL18 in three fractions from the 60-80S region

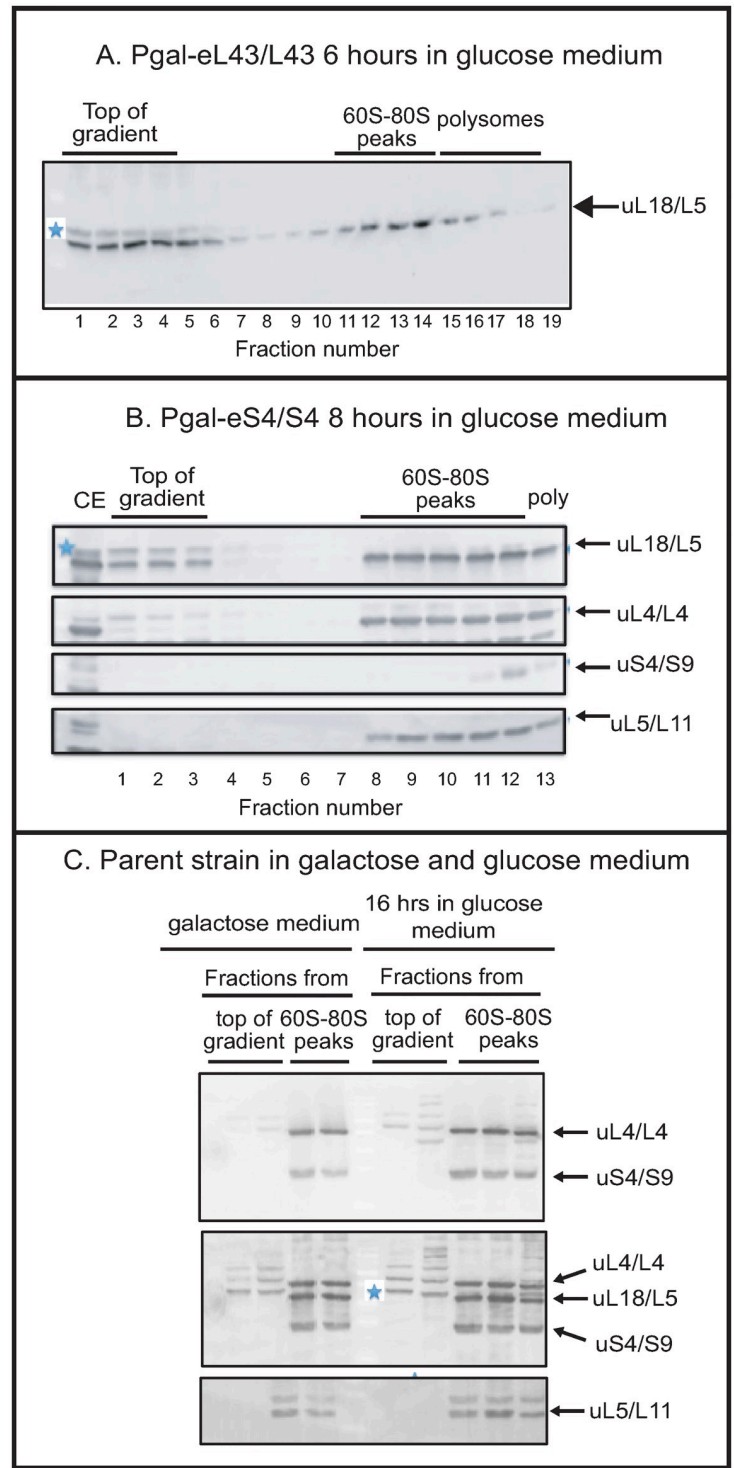

**Fig 2. Repression of the gene for the 40S r-protein eS4/S4 causes accumulation of extra-ribosomal uL18/L5, but not extra-ribosomal uL5/L11, uL4/L4, or uS4/S9.** $P_{gal}$-eL43/L43, $P_{gal}$-eS4/S4, and the parent strain BY4741 were grown in galactose medium and switched to glucose medium for the indicated time. Whole-cell lysates were fractionated on sucrose gradients and the indicated fractions of the gradients were analyzed by western blots probed with antisera for r-proteins uL18/L5, uL4/L4, eS4/S4, and uL5/L11 as indicated at each blot. (A) $P_{gal}$-eL43/L43 after 6 hours in glucose medium. All fractions from the top of the gradient through the polysome region are shown. (B) $P_{gal}$-eS4/S4 after 8 hours in glucose medium. The panel shows sections cropped from a western blot loaded with aliquots of each fraction from the top of the gradient through the polysome region. The blot was first probed with anti-uL18/L5,

then with a mixture of antisera for uS4/S9 and uL4/L4 without stripping. Finally, the bottom of the blot was probed with anti-uL5/L11. (C) The parent strain (BY4741) after 0 and 16 hours in glucose. Aliquots of consecutive fractions from the top of the gradient and the 60S-80S region were analyzed by western blot. The top panel shows the blot probed with a mixture of antisera for uL4/L4 and uS4/S9. The middle panel shows the same blot after it was probed further with anti-uL18/L5 without stripping. The bottom panel shows the same blot after it was probed further with anti-uL5/L11. Uncropped images are shown in S1 Fig. The bands marked with a blue star in some panels are not related to uL18/L5 (see Fig 1 in Material and Methods). CE: cell extract.

(S2 Table). The results show that there is a good semi-quantitative agreement between the two approaches. Hence, we conclude that it is justified to use only selected fractions for comparing the amount of extra-ribosomal uL18 in different strains.

## Protein molecular weight markers

Protein molecular weight markers were purchased from Thermo Fisher Scientific (cat 26616). Note that r-proteins always run slower in SDS gels than expected from their molecular weight.

## Statistics

Pairwise t-test was used.

# Results

## Disruption of ribosome assembly

To specifically abolish the synthesis of individual essential r-proteins cognate to one or the other ribosomal subunit in *S. cerevisiae*, we used yeast strains in which the only gene for a given r-protein is transcribed from the *GAL1/10* promoter. We refer to these strains as $P_{gal}$-xx, where xx is the name of the protein encoded by the gene under galactose control. In galactose medium, a full set of r-proteins is synthesized, but shifting the cells to glucose medium abrogates the synthesis of r-protein xx, which prevents assembly of the cognate ribosomal subunit [11].

## Extra-ribosomal uL18 accumulates during repression of some 40S r-protein genes

To measure extra-ribosomal accumulation of uL18 and several other r-proteins upon repression of specific r-protein genes, we fractionated crude cell extracts on sucrose gradients and analyzed the sucrose gradient fractions on western blots probed with antisera specific to the 60S r-proteins uL18, uL5, uL4 and the 40S r-protein uS4. Fig 2A shows a western blot stained with anti-uL18 of fractions from a sucrose gradient loaded with an extract of $P_{gal}$-eL43 prepared 6 hours after shifting the culture from galactose to glucose medium. A band co-migrating with the ribosomal uL18 band in the gel was observed close to the top of the sucrose gradient. A second protein marked with a star and moving slightly slower also appeared. As described in Materials and Methods we confirmed that the protein that co-migrates with the ribosomal uL18 band indeed represents uL18, while the slightly slower moving protein is not related to uL18 (Fig 1). The appearance of uL18 at the top of the gradient after repressing the eL43 gene was confirmed by transforming $P_{gal}$-eL43 with a plasmid carrying a constitutive gene for FLAG-tagged uL18. As seen in Fig 1B, the uL18-FLAG accumulated at the top of the gradient after the shift to glucose medium, but not before the shift (Fig 1B). Thus, the experiments in Figs 2A and 1B show that uL18 accrues outside ribosomal particles during the repression of uL43 synthesis. An extra-ribosomal pool of uL18 was also seen after repressing uL5 synthesis; see note about nomenclature in Materials and Methods [14].

To determine if abolishing expression of 40S r-proteins also triggered extra-ribosomal uL18 accumulation, we repressed the synthesis of eS4, a protein incorporated into the 40S precursor particle (pre-40S) early in the assembly process [19]. After 8 hours of repression, extracts were analyzed by sucrose gradient centrifugation and western blot. Unexpectedly, we found a build-up of extra-ribosomal uL18 at the top of the sucrose gradient, but no uL4, uL5 or uS4 was found outside of the ribosome peaks (Fig 2B). Additionally, the parent strain BY4741 did not accumulate extra-ribosomal r-proteins whether grown in galactose or glucose, as expected since assembly of both subunits proceeds uninterrupted in the parent whether it grows in galactose or glucose medium (Fig 2C). Overall, the results in Fig 2 show that repression of the eS4 gene with ensuing disruption of 40S assembly generates a pool of extra-ribosomal uL18. Extra-ribosomal accumulation of uL18 is thus not specific to interference with 60S assembly.

We then tested if repression of the genes for 40S r-protein genes eS6, uS17, and eS19 also cause accumulation of extra-ribosomal uL18. The eS31 protein was chosen because it is incorporated into the pre-40S late in the assembly process [20] (as opposed to eS4, which is incorporated early), and proteins eS6, uS17, eS19 were chosen because mutations in the orthologous human genes have been implicated as causing diseases [21–24]. Repression of eS4, eL40, and eL43 was used as controls. Since 60S particles are largely stable after repression 40S r-protein genes and the cultures grow at somewhat different rates after the shift to glucose [11], we reasoned that quantifications of extra-ribosomal uL18 would be more comparable if the density of all cultures were allowed to increase by the same factor after the shift to glucose medium. Accordingly, we harvested the cultures after the $OD_{600}$ had increased at least four-fold during incubation in glucose rather than at a specific time after the change of carbon source. As shown in Fig 3A we found uL18 at the top of the gradient for all strains, except $P_{gal}$-eS31, but the strength of the bands varied. To estimate the fraction of uL18 present at the top of the sucrose gradient we quantified uL18 in two top fractions from the sucrose gradient and three fractions in the 60S-80S ribosome peaks (see Methods and materials for a justification of this approach). The results showed that repression of eS4 synthesis generated approximately as much extra-ribosomal uL18 as did the repression of the two 60S r-protein genes while repressing other 40S r-protein genes generated smaller amounts of extra-ribosomal uL18 (Fig 3B). Furthermore, the amount of extra-ribosomal uL18 differed significantly after repressing the various 40S r-protein genes (Fig 3C).

## The extra-ribosomal uL18 pool reaches a steady-state

Since the pool size of the extra-ribosomal uL18 pool differed between strains (Fig 3B), we questioned whether the pool changes with time. Accordingly, we compared the uL18 pools in $P_{gal}$-eL43, -uS4, and -eS31 harvested at different times after the shift to glucose medium. The extra-ribosomal uL18 pool increases by about 2-fold between 4 and 6 hours of repression of either eS4 and eL43, but is constant between 6 and 8 hours, suggesting that the extra-ribosomal uL18 pool reaches a steady-state (Fig 4A and 4B) by the time the cultures have gone through about two doublings after the shift to glucose medium. Furthermore, the pool of extra-ribosomal uL18 does not rise above the statistically significant level even after 17 hours of repressing eS31 synthesis (Fig 4C).

## uL18 accrues due to interference with subunit assembly, not degradation of mature subunits

**Maintenance of the extra-ribosomal uL18 pool requires protein synthesis.** We have previously shown that 60S formation continues during the repression of 40S assembly and slow turnover of the 60S begins a few hours after 40S r-proteins have been repressed [11].

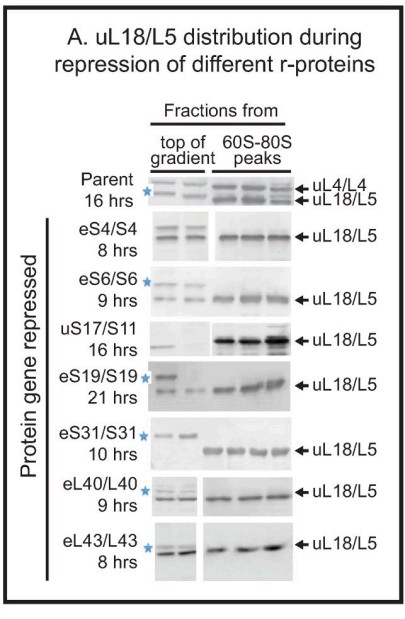

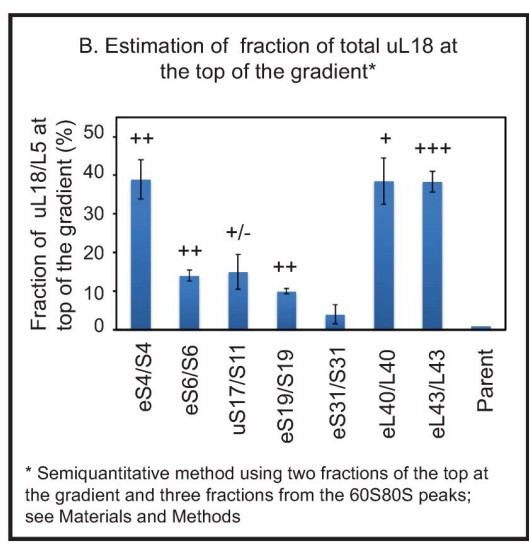

**Fig 3. Quantification of extra-ribosomal uL18/L5 pool after repression of different 40S and 60S r-protein genes.**
$P_{gal}$-eS4/S4, -eS6/S6, -uS17/S11, -eS19/S19, -eS31/S31, -eL40/L40, -eL43/L43, and the parent strain BY4741 were grown in galactose medium and shifted to glucose medium. (A) Sucrose gradients were loaded with lysates prepared after at least a four-fold increase in $OD_{600}$ after the shift to glucose medium, at which time the extra-ribosomal pool of uL18 had reached equilibrium (see Fig 4). Aliquots of consecutive fractions from the top and the 60S-80S region of the gradient were quantified from western blots probed with anti-uL18/L5. The protein whose synthesis was repressed by the shift to glucose medium is shown on the left and actual times in glucose medium for each strain are indicated below the protein name. For the abolition of the synthesis of eS4/S4, uS17/S11, e40/L40, and eL43/L43, the image was cropped from blots that include all fractions from the sucrose gradient. For repression of eS6/S6, eS19/S19, and eS31/S31 only aliquots from the top fractions and the 60S-80S fractions were analyzed by western blot, but see also Fig 4C for a parallel experiment with Pgal-eS31/S31 in which all fractions were included in the western blot. Uncropped images are shown in Fig 2A and 2B and S1, and S2 Figs. (B) Quantification of uL18 at the top of the sucrose gradient after repressing different r-protein genes. The blots in Panel A were quantified using ImageJ. The sum of uL18 in the two top fractions was normalized to the sum of uL18 found in all five fractions. This procedure for estimating the fraction of total uL18 at the top of the gradient is validated in Materials and Methods (S2 Table). The average of three biological repeats for eS4/S4, uS17/S11, eS31/S31, eL43/L43 and two biological replicates eS6/S6, eS19/S19, eL40/L40 is shown together with the standard error of the mean. The data for each gene repression experiment was compared to the results from the parent strain by pairwise t-test. +++ indicates p<0.001, ++ p<0.005, + p<0.01, +/- p<0.1. (C) T-tests for pairwise comparison of results from different strains.

Thus, there are two possible principle sources of extra-ribosomal uL18: modification of 60S assembly and breakdown of mature 60S subunits. To distinguish these possibilities, we investigated if blocking protein synthesis with cycloheximide change the amount of extra-ribosomal uL18. If the extra-ribosomal uL18 stems from degradation of preexisting ribosomes, cycloheximide should not affect the pool of extra-ribosomal uL18, but if the extra-ribosomal uL18 is generated in the assembly process, the formation depends on continual protein synthesis and addition of cycloheximide should reduce the pool of extra-ribosomal uL18. Hence, we grew $P_{gal}$- eL43 in galactose and shifted it to glucose medium for 6 hours. At this time approximately 50% of the total uL18 was found at the top of the gradient (Fig 4A(i)). The culture was then split and cycloheximide was added to one aliquot to inhibit total protein synthesis, while

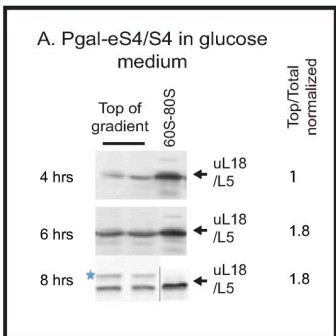

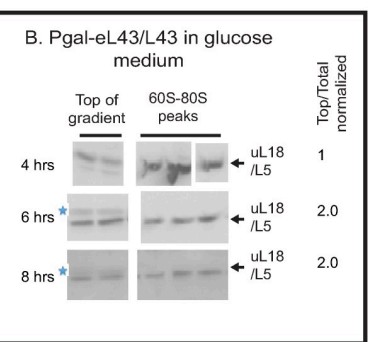

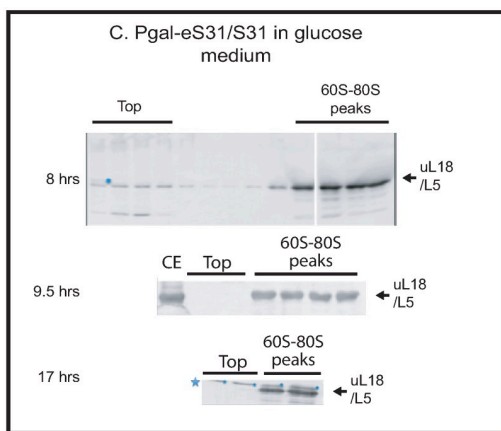

**Fig 4. Extra-ribosomal uL18 as a function of time after repressing eS4, eL43, and eS31.** $P_{gal}$-eS4, -eL43 and -eS31 were grown in galactose medium and shifted to glucose medium. At the indicated times after the shift, extra-ribosomal uL18 was quantified. (A) $P_{gal}$-eS4 and (B) $P_{gal}$-eL43. Consecutive fractions from the top and the 60S-80S region of the gradient were analyzed. The blots were scanned and quantified using Photoshop CC 2019. The ratio of uL18 found in the top fractions relative to uL18 in all fractions on the western blot was calculated for each time point and normalized to the value for the 4-hour sample. (C) $P_{gal}$-eS31. All fractions from the top of the sucrose gradient through the polysome region are shown for the 8-hour sample. For the 9.5 and 17-hr samples, consecutive fractions from the top and the 60S-80S region of the sucrose gradient are shown. See S3 Fig for uncropped blots. CE cell extract.

nothing was added to the other part. After 4 hours of additional culturing, both aliquots were harvested and analyzed for extra-ribosomal uL18. No uL18 band was seen at the top of the gradient after cycloheximide inhibition of protein synthesis (Fig 4A(ii)), while the level of extra-ribosomal uL18 was unchanged in the sample without the drug (Fig 4A(iii)).

Inhibition of protein synthesis in $P_{gal}$-eS4 gave a similar result. Cycloheximide was administrated for 15 and 45 minutes to a culture four hours after the shift from galactose to glucose. While no change was seen after 15 minutes, the extra-ribosomal uL18 level was reduced by about 50% after 45 minutes with cycloheximide (Fig 5B). Together the experiments in Fig 5A and 5B show that the extra-ribosomal uL18 is depleted, if it is not replenished by new synthesis, indicating that extra-ribosomal uL18 is generated during 60S assembly rather than degradation of mature 60S subunits.

**No extra-ribosomal uL18 accumulates during degradation of cytoplasmic ribosomes.** To determine if extra-ribosomal uL18 also accumulates during degradation of cytoplasmic ribosomes, we inhibited TOR-activity by rapamycin. This stops rRNA synthesis and causes a significant degradation of cytoplasmic ribosomes [25]. Rapamycin (0.2 μg/ml) was added to Pgal-eL43 in growing in **galactose** medium (i.e. eL43 synthesis is **not** interrupted) for 4 hours before a lysate was analyzed by sucrose gradient centrifugation and western blot. As seen in Fig 6, no uL18 was seen at the top of the gradient after 6 hours. Since there is significant cytoplasmic degradation of ribosomes during rapamycin inhibition of TOR [25], extra-ribosomal uL18 buildup would have been expected, if it comes from ribosome degradation. We did not see that, suggesting that the degradation of mature ribosomes does not generate extra-ribosomal uL18.

## Stabilization of extra-ribosomal uL18 does not require Rrs1 or Rpf2

We further investigated the origin of extra-ribosomal r-proteins by depleting each of the ribosomal assembly factors Rrs1 and Rpf2 that combine with uL18, uL5, and 5S rRNA before docking in the precursor 60S particle (pre-60S) [26]. If the protection of uL18 from rapid degradation requires formation of the full pre-docking complex, no extra-ribosomal uL18 should

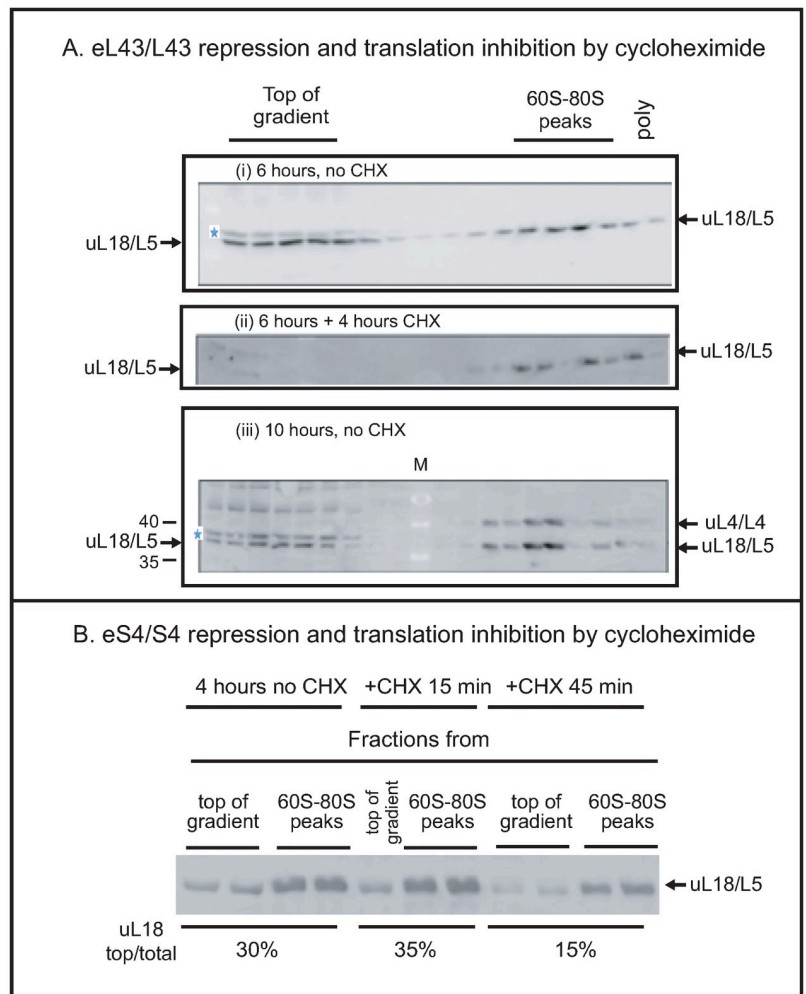

**Fig 5. Maintenance of the extra-ribosomal uL18 pool requires protein synthesis.** (A) Cycloheximide decreases the pool of extra-ribosomal uL18/L5 during the repression of a 60S r-protein gene. Six hours after shifting $P_{gal}$-eL43/L43 from galactose to glucose medium, cycloheximide was added to an aliquot of the culture (final concentration: 100 μg/ml), while the culturing of another aliquot of the culture was continued without addition of the drug. Both aliquots were harvested 10 hours after the shift. Whole-cell extracts were analyzed by sucrose gradients and western blots of aliquots of all sucrose gradient fractions. (i) $P_{gal}$-eL43/L43 6 hours after the shift of media, (ii) Pgal-eL43/L43 incubated with cycloheximide added 6 hours after the media shift and harvested 10 hours after the shift. (iii) $P_{gal}$-eL43/L43 incubated for 10 hours in glucose without cycloheximide. All fractions from the top through the polysome region of the sucrose gradient are shown. (B) Cycloheximide also decreases the pool of extra-ribosomal uL18/L5 during the repression of a 40S r-protein gene. $P_{gal}$-eS4/S4 was grown in galactose medium and shifted to glucose medium. Cycloheximide (100 μg/ml) was added to the culture 6 hours after the shift and cells were harvested 0, 15 and 45 minutes after addition of the drug. Aliquots of one or two fractions from the top of the gradient and two fractions from the 60S-80S peaks were analyzed by western blots probed with anti-uL18/L5. Consecutive fractions from the top and the 60S-80S region of the sucrose gradient were shown. M Molecular weight markers in kD. 35 and 40 kD are identified. See S2 Fig for the full molecular weight ladder.

accumulate during inhibition of the two assembly factors. However, Fig 7 shows that depleting either Rrs1 or Rpf2 increased the pool of extra-ribosomal uL18 in agreement with the effect of mutating the *RRS1* gene [27]. This shows that stabilization of extra-ribosomal uL18 does not require the formation of the complete uL18-uL5-5S rRNA-Rrs1-Rpf2 particle from. This is also supported by the fact that extra-ribosomal uL5 does not accumulate proportionally to uL18 during abrogation of eS4 or eL40 synthesis (Fig 2B and S2 Fig). Furthermore, the buildup

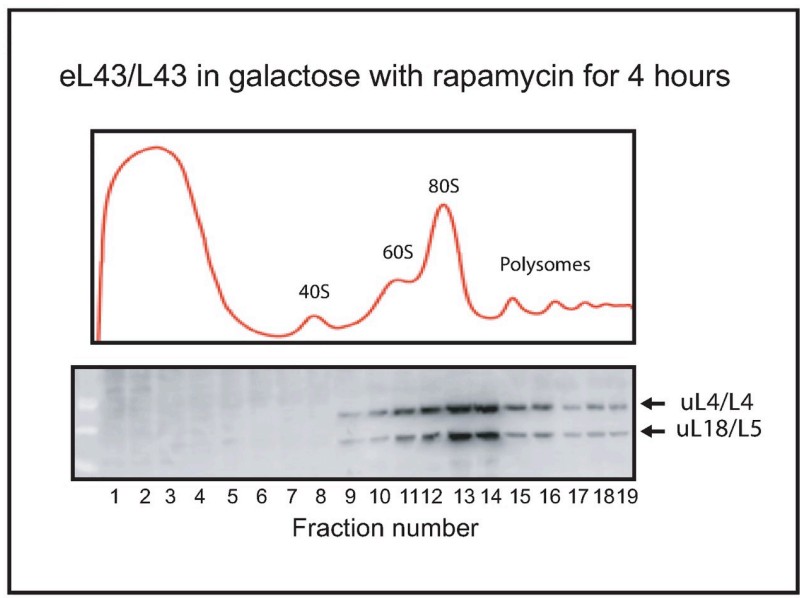

**Fig 6. Extra-ribosomal uL18 does not accumulate during degradation of cytoplasmic ribosomes.** Rapamycin was added to Pgal-eL43/L43 growing in **galactose** medium (**no** shift to glucose) and 4 hours later an extract was fractionated on a sucrose gradient centrifugation and aliquots of fractions were analyzed on a western blot developed with antisera specific to uL18/L5 and uL4/L4. All fractions from the top through the polysome region of the sucrose gradient are shown.

of extra-ribosomal uL18 concurs with the conclusion that accrual of extra-ribosomal uL18 stems from the 60S assembly process.

## Discussion

### Inhibition of 40S assembly changes 60S assembly kinetics

Ribosomal proteins are rapidly turned over unless they are incorporated into ribosomes [10, 28, 29]. However, uL18 escapes this degradation and accumulates in extra-ribosomal complex

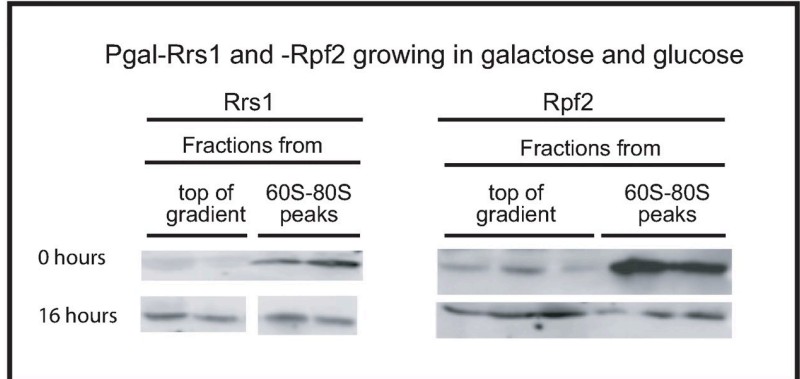

**Fig 7. Accumulation of extra-ribosomal uL18 does not require formation of the complete uL18-uL5-5SrRNA-Rrs1-Rpf2 complex.** $P_{gal}$-Rrs1 and -Rpf2 were grown in galactose medium and shifted to glucose medium for 16 hours. Whole-cell extracts were analyzed by sucrose gradient centrifugation and aliquots of two or three fractions from the top of the gradient and two fractions from the 60S-80S peaks were analyzed by western blots developed with antiserum specific to uL18/L5. Consecutive fractions from the top and the 60S-80S region of the sucrose gradient were shown.

(es) when the synthesis of 60S r-proteins is curtailed as shown in Fig 2 and [14]. Here we have shown that extra-ribosomal uL18 also builds up during repression of several 40S r-protein genes (Figs 2 and 3). This was unexpected because abrogation of 40S r-protein synthesis specifically halts the assembly of the 40S subunit, while assembly of the 60S subunit continues [11]. Thus, the build-up of the extra-ribosomal uL18 pool reveals a novel form of interaction between 40S and 60S subunit formation: abolition of 40S subunit synthesis changes the kinetics of the individual steps in 60S subunit assembly, but has little effect on the overall rate of 60S formation.

Several of observations indicate that extra-ribosomal uL18 comes from the 60S assembly process rather that degradation of mature cytoplasmic 60S subunits. First, maintenance of the pool of extra-ribosomal uL18 requires continual protein synthesis, whether provoked by disruption of the formation of the 60S or the 40S subunit (Fig 5A and 5B). Second, no extra-ribosomal ul18 is seen during administration of the TOR inhibitor rapamycin (Fig 6), which causes significant degradation of cytoplasmic ribosomes [25]. Third, depletion of either of two nuclear 60S ribosome biogenesis factors leads to a buildup of extra-ribosomal uL18 (Fig 7). The formation of extra-ribosomal uL18 during 60S assembly also concords with the mapping of extra-ribosomal uL18 to the nucleus where most ribosome formation takes place [14].

The docking of uL18 into the nascent 60S subunit involves the formation of a uL18-uL5-5S rRNA-Rrs1-Rpf2 complex [26], but, interestingly, two observations show that the protection of extra-ribosomal uL18 against rapid turnover does not require the formation of this complex in its entirety. This conclusion is based on three findings: First, here we show that repression of the Rrs1 or Rpf2 genes cause the buildup of an extra-ribosomal uL18 pool (Fig 7). Second, extra-ribosomal uL5 does not accrue in parallel with extra-ribosomal uL18 (Fig 2B and S2 Fig). In agreement with this, experiments published by the Woolford lab showed that extra-ribosomal uL18 accumulates when the uL5 (alias L11) synthesis is abolished [14] [Note that uL5/L11 was named L16 when this paper was published.].

## Co-assembly of 40S and 60S precursor ribosomes may account for the interaction between the ribosomal assembly processes

Our results can be interpreted mechanistically in the context of current models for rRNA processing and ribosome assembly. During rapid growth of yeast cells, the nascent RNA Polymerase I pre- rRNA is cleaved at the A2 site in the Internal Transcribed spacer 1 (ITS1; Fig 8A) when the RNA polymerase has transcribed about half of the 60S-specific sequences (co-transcriptional pre-rRNA cleavage) [30]. Furthermore, several steps towards forming the ribosomal precursor particles occur before pre-rRNA cleavage, including significant pre 40S compaction and emergent pre-60S compaction [30] and association of many ribosomal assembly factors and at least some r-proteins with the pre-rRNA [19, 31] (Fig 8B(i)). If the formation of the pre-40S (90S) is inhibited, the cleavage of the pre-rRNA is delayed until transcription is completed (post-transcriptional cleavage) and there is essentially no compaction of either the pre-40S or pre-60S (Fig 8C(i)) [30, 32–34].

To explain our results within this framework, we propose that the co-transcriptional compaction and assembly of the pre-40S during uninhibited (fast) growth promotes the folding the 5' part of the 60S rRNA, early assembly of the pre-60S, and the co-transcriptional cleavage. Since interference with 40S assembly prevents both co-transcriptional cleavage and 40S compaction [30] (Fig 8C(i)), this implies that the rRNA folding and pre-60S structure is different during co- and post-transcriptional cleavage.

We further posit that the pre-60S formed during post-transcriptional processing has a lower affinity for uL18-5SrRNA complex than does the pre-60S formed during the co-transcriptional

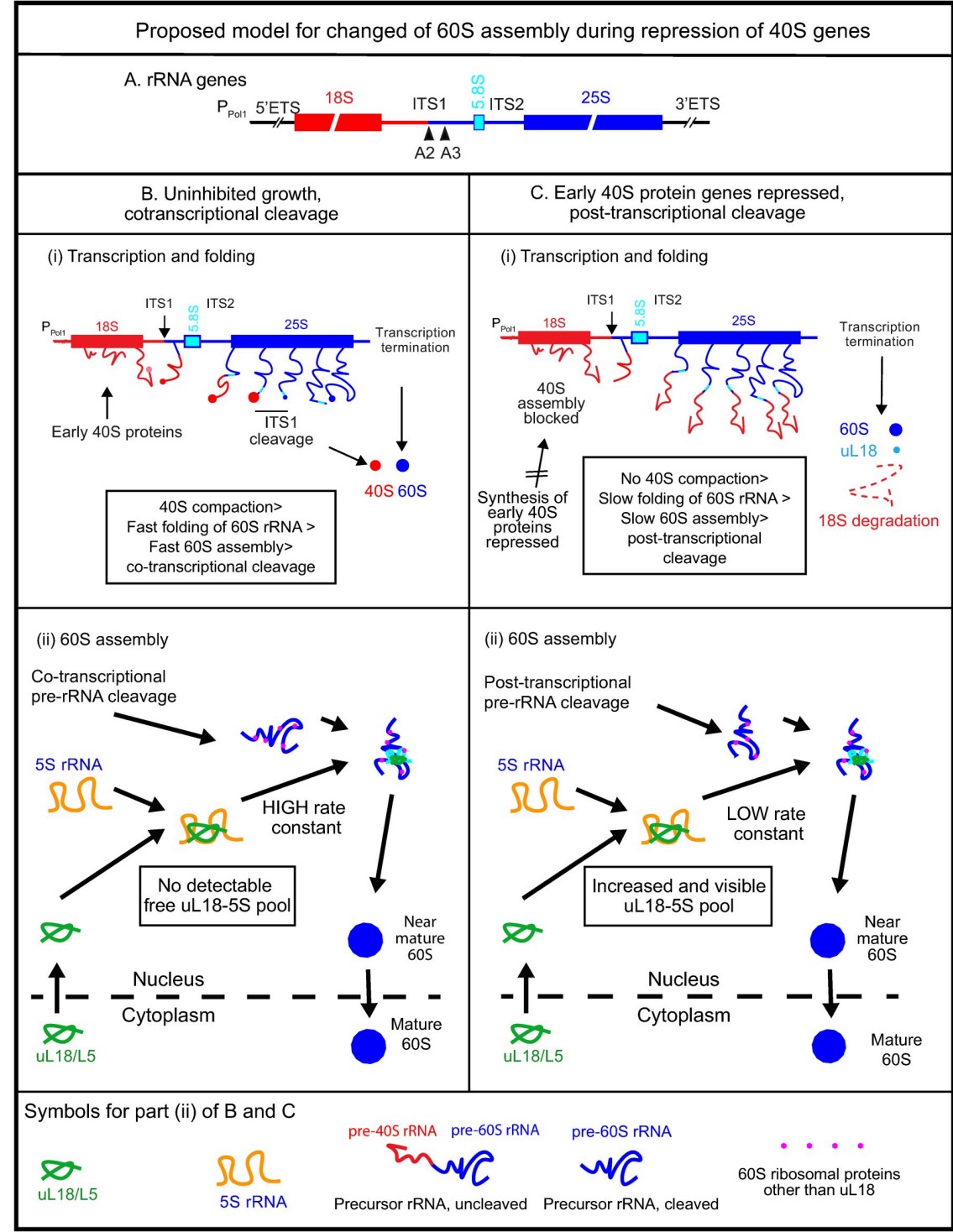

**Fig 8. Model for effect of early 40S assembly on the kinetics of 60S assembly.** (A) Map of the long rRNA precursor gene transcribed by RNA polymerase I. ETS: External Transcribed Spacer; ITS: Internal Transcribed Spacer; A2 cleavage site separating the pre-rRNA into subunit-specific parts. (B) Co-transcriptional rRNA cleavage during unrestricted 40S assembly. (i) Schematic representation of co-transcriptional rRNA cleavage and ribosomal assembly as revealed by electron micrographs (Miller Spreads) [30]. The 18S parts of the transcript is in red, 5.8S in turquoise, and 25S in blue. "Wiggled" lines: uncompacted rRNA complexed with assembly factors and early assembly r-proteins. Filled circles: Compacted rRNA-protein complexes; the intensity of the red color indicates the density (compaction) of the complex. (ii) Kinetic model for formation of the uL18-5S rRNA complex and incorporation into the pre-60S during unrestrained 40S assembly. We propose that the rate constant for uL18/5S rRNA binding to the pre-60S is

high in this condition. Therefore, uL18 is rapidly incorporated into the pre- 60 and uL18 does not accumulate outside the ribosome. (C) Post-transcriptional cleavage during inhibited 40S assembly. (i) Schematic illustration of electron micrographs of rRNA genes during repression of 40S r-protein genes. Pre-rRNA cleavage happens after transcription termination and compaction of the pre-40S and pre-60S is essentially absent. We suggest that absence of 40S compaction delays cleavage and changes the folding of the 60S-specific rRNA. (ii) Kinetic model for formation of the uL18-5S complex and incorporation into the pre-60S. We propose that the affinity (rate constant) of uL18-5S rRNA binding to the pre-60S is reduced due to the alternate folding and, possibly, alternate assembly pathway during post-transcriptional cleavage. Therefore, the rate of incorporation of uL18 into the pre-60S is slow leading to formation of an extra-ribosomal uL18 pool.

cleavage pathway, implying that the rate constant for binding of the uL18-5S rRNA to pre-60S is lower if the pre rRNA is cleaved after transcription termination. Therefore, the switch from co-transcriptional to post-transcriptional cleavage initially reduces the rate of the incorporation of the uL18-5S rRNA complex into the pre-60S. Since the rate of r-protein and rRNA synthesis is not reduced by the inhibition of assembly, the concentration of extra-ribosomal uL18-5S rRNA increases, causing the rate of uL18-5S rRNA incorporation into the pre-60S to rebound (rate = rate-constant x concentration of uL18-5S rRNA) in accordance with the continued assembly of the 60S subunit. Overall, our model thus explains why the uL18 accumulates outside of the ribosome during post-transcriptional cleavage, while 60S assembly continues relatively unabated. This bears resemblance to changes in the kinetics of 60S assembly provoked by mutations in *Escherichia coli* uL4. In this case, the mutations slow down one or more steps in the 60S assembly pathway causing a rise in 60S assembly intermediates, visible as new peaks in sucrose gradients, while the formation of mature 60S subunits continues [35].

Whereas we think that our proposed model is likely to be part of the mechanism for regulating the uL18 extra-ribosomal pool, other factors may also change the 60S structuring process and the kinetics of 60S assembly. For example, depletion of either RNase MRP (cleaving at the A3 site in ITS1), Rat1 5'>3' exonuclease, and depletion of nutrients affect the balance between co- and post transcriptional cleavage [30, 32, 33]. However, the effect of such parameters on the accumulation of extra-ribosomal r-proteins is not known at the present time.

Interactions between 40S and 60S formation is likely stronger in metazoans than in yeast, because a larger fraction of pre-rRNA is cleaved into subunit-specific pieces after completion of transcription ("post-transcriptionally rRNA processing") in metazoans than in fast-growing yeast cells. The difference between the ratio of co-transcriptional and post-transcriptional is evident from Northern blots of rRNA processing intermediates in the two types of organisms: full length rRNA precursor transcript is more prevalent relative to other processing intermediates in mammalian cells (e.g. [36]) than it is in fast-growing yeast cells (e.g. [11]).

While the vast majority of the r-protein mass is tied up in physical ribosomes, extra-ribosomal r-proteins are thought to be involved in regulation of cell growth and other functions, at least in metazoan cells. For example, r-proteins from both ribosomal subunits have been identified as cancer drivers [37]. The mechanism for r-protein-mediated regulation of growth and cell fate presumably involves binding of r-proteins to regulators of growth and the progression of the cell cycle [38–41]. The functions of extra-ribosomal proteins have been intensely investigated, but little is known about the origin of the extra-ribosomal r-protein pools. Since the major features of pathways for ribosomal assembly are evolutionarily conserved, we suggest that our analysis in the yeast model organism also contributes to understanding the complexity of how ribosome assembly impacts regulation of growth in metazoan cells.

## Supporting information

**S1 Fig. Uncropped versions of images in Fig 2A and 2B.**
(PDF)

**S2 Fig. Uncropped versions of images in Fig 3A.** CE whole cell extract. M molecular weight markers in kD.
(PDF)

**S3 Fig. Uncropped versions of images in Fig 4.**
(PDF)

**S1 Table. Stains used.**
(XLSX)

**S2 Table. Quantification of extra-ribosomal uL18.** The fraction of uL18 at the top of the sucrose gradient was determined by (i) quantifying uL18 in all fractions, or (ii) quantifying uL18 in two top fractions and three fractions from the 60S-80S peaks. Photoshop CC 2019 was used for the quantification.
(XLSX)

# Acknowledgments

We thank Drs. Philipp Milkereit (University of Regensburg, Germany) and John Woolford (Carnegie Mellon University, Pennsylvania, USA) for strains and plasmids. We also thank Benedikte Traasdahl for help with the manuscript.

# Author Contributions

**Conceptualization:** Nusrat Rahman, Md Shamsuzzaman.

**Data curation:** Nusrat Rahman.

**Formal analysis:** Nusrat Rahman, Md Shamsuzzaman, Lasse Lindahl.

**Funding acquisition:** Lasse Lindahl.

**Investigation:** Nusrat Rahman.

**Methodology:** Nusrat Rahman, Md Shamsuzzaman.

**Resources:** Lasse Lindahl.

**Supervision:** Lasse Lindahl.

**Visualization:** Nusrat Rahman, Lasse Lindahl.

**Writing – original draft:** Nusrat Rahman, Lasse Lindahl.

**Writing – review & editing:** Nusrat Rahman, Md Shamsuzzaman, Lasse Lindahl.

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
