## [Decision Letter · Decision Letter 0]

20 Sep 2019

PONE-D-19-24366

Interaction between the assembly of the ribosomal subunits: Disruption of 40S ribosomal assembly causes accumulation of extra-ribosomal 60S ribosomal protein uL18/L5

PLOS ONE

Dear Lasse,

Thank you for submitting your manuscript to PLOS ONE. Your manuscript was reviewed by two experts in the ribosome biogenesis field. As you can see from the reviews, their comments were generally very positive, however, they also highlighted a number of inconsistencies that need further clarification. In particular Reviewer #2 raised a number of important points regarding the quantification of the data as well as comparison with previously published work. Therefore, we invite you to submit a revised version of the manuscript that addresses the points raised during the review process.

We would appreciate receiving your revised manuscript by Nov 04 2019 11:59PM. To enhance the reproducibility of your results, we recommend that if applicable you deposit your laboratory protocols in protocols.io, where a protocol can be assigned its own identifier (DOI) such that it can be cited independently in the future. For instructions see: http://journals.plos.org/plosone/s/submission-guidelines#loc-laboratory-protocols

We look forward to receiving your revised manuscript.

Kind regards,

Sander Granneman

Academic Editor

PLOS ONE

Journal Requirements:

"This work was supported by grants from the National Science Foundation, USA (0920578), The Benelein Technologies, LLC, and the University of Maryland, Baltimore County. We thank Drs. Philipp Milkereit (University of Regensburg, Germany) and John Woolford (Carnegie Mellon University, Pennsylvania, USA) for strains and plasmids. We also thank Benedikte Traasdahl for help with the manuscript."

"The funders had no role in study design, data collection and analysis, decision to publish, or preparation of the manuscript"

Reviewers' comments:

Reviewer's Responses to Questions

**Comments to the Author**

1. Is the manuscript technically sound, and do the data support the conclusions?

Reviewer #1: Yes

Reviewer #2: Partly

2. Has the statistical analysis been performed appropriately and rigorously? 

Reviewer #1: Yes

Reviewer #2: I Don't Know

3. Have the authors made all data underlying the findings in their manuscript fully available?

Reviewer #1: Yes

Reviewer #2: Yes

4. Is the manuscript presented in an intelligible fashion and written in standard English?

Reviewer #1: No

Reviewer #2: Yes

5. Review Comments to the Author

Reviewer #1: This manuscript by Rahman, Shamsuzzaman and Lindahl is an interesting follow-up to a previous paper from the Lindahl group showing interdependence of assembly of large and small ribosomal subunits in yeast. Here they demonstrate that ribosomal protein uL18, aka L5, accumulates outside of ribosomes when assembly of either large or small subunits is blocked by depleting a ribosomal protein from one or the other subunit. This was visualized by gradient sedimentation of L5 in extra-ribosomal fractions. Of interest is how varying effects of depletion of different r proteins reflects the order and mechanism of protein and subunit assembly, including effects resulting from post-transcriptional assembly.

The conclusions are sound and the results are interesting. However, the story might be more impactful if the following were added:

(1 Include a figure showing in more detail which SSU and LSU r proteins are assembled early and which much later, and thus why depletion of some might have a greater effect on L5. This is explained in the text but would be clearer to nonaficionados if a figure/cartoon were added.

(2 L5 accumulates in fractions at the top of the gradient. Are 5S rRNA, L11, Rpf2 and Rrs1 also present in the same complex with L5? CO-IP from gradient fractions will demonstrate whether this is the case. This is potentially important since the authors report accumulation of extraribosomal L5 when Rpf2 or Rrs1 are depleted (Fig.4D).

Minor issues

(1 Some figures need to be better labeled and there are numerous places where a word or two need to be added or changed.

Line 56: Aren’t some r proteins not essential, i.e. not completely required for subunit assembly?

Line 119 and elsewhere: a detail perhaps: is the symbol in the figures a star or an asterisk?

Line 151 and elsewhere: the term GAL1/10 promoter should be capitalized and italicized.

Line187: what is meant by Northern blots of fractions were probed? Don’t you mean just “blots”?

Line 367: is ther a word or two missing?

Figures: Does more than one lane of blot data correspond to consecutive gradient fractions, e.g. lanes 3,4 and 5,6 in Fig.1? If so, please explain in the figure legends.

Figure 2B: the text states that L11 was assayed, but it is not shown.

(2 In the Introduction, lines 64-66, the authors need to correct their description of the experiment done by Dehmukh et al. They got trapped by the nomenclature monster! R protein L16 was depleted, which is NOT the same as uL5, as stated. This is important because uL5 aka L11 is in an assembly subcomplex with uL18 aka L5, whereas L16 is not.

Reviewer #2: Rahman et al. report in the manuscript PONE-D-19-24366 on the results of a series of experiments which provide evidence for the extra-ribosomal accumulation of the large ribosomal subunit protein uL18 upon inhibition of small ribosomal subunit assembly in S. cerevisiae. Their conclusions are based on quantitative interpretation of western blotting analysis after sucrose gradient fractionation of cellular extracts prepared from yeast ribosomal protein expression mutants.

In this regard the exact approach chosen might be better described. The authors refer to reference 9, in which an indirect immuno-detection is described using an enzymatic reaction for secondary antibody visualisation. These detection approaches tend to produce saturation artefacts and it might be therefor important to know whether titration of a reference sample was used in the individual western blots to determine the dynamic range of the detected signal. For the sucrose gradient fractionation, in several experiments all fractions were analysed. In these cases the ribosomal proteins could be detected in four to six of the “Top” fractions and in up to eleven of the faster sedimenting “Ribosomes” fractions. In several other analyses only one or two of the “Top” fractions and either two or three of the “Ribosomes” fractions of the sucrose gradients were analysed to deduce the ratio of extra-ribosomal to ribosomal r-proteins as uL18. How was it ensured that these fractions contain all of the extra-ribosomal and ribosomal pools of uL18, or are representative for them?

The western blot results shown in Fig 3A , 3C(iii) and 4D look quite cropped. That complicates interpretation of the data, especially since the uL18/L5 antibody used in this study cross-reacts with a protein running just a bit slower than uL18/L5. The identity and running behavior of the band detected in the “Top” fractions in 3A for the parental strain is, for example, in the actual representation for me ambiguous. Thus, less cropped figures might help the reader in these cases.

For several of the experiments (Fig 3A +B) I could not find the exact incubation times in glucose containing medium which were applied to repress the expression of individual r-proteins. In the materials and methods section it is stated that cultures were shifted for 6 to 21 hours to glucose containing medium. In one experiment (Fig. 2) the control strain is shifted to glucose containing medium for a different time period (16 hours) than the two r-protein expression mutants analysed (6 hours and 8 hours). Why was that done, and can we be sure that this might not affect the outcome of the experiments? At least from the data shown in Fig 3C for Pgal-eS31 I got the impression that the level of free uL18/L5 decreases with prolonged incubation times in glucose containing medium.

Recently, changes in levels of proteins in several large and small subunit r-protein expression mutants were measured by mass spectrometry proteome wide (https://doi.org/10.1016/j.molcel.2018.10.032). Did the authors analyse these published datasets for a possible excess of uL18/L5 over other large subunits proteins as ul4/L4 or uL5/L11? That might give further support to the authors conclusions by an analysis performed with an independent experimental approach.

A few other comments and remarks:

1) Line 52: “..also requires in excess of 250 ribosomal assembly factors”

Does that refer to the stochiometry of components?

2) Line 54: “..while others are specific to the formation of one the ribosomal subunits”

of one of ?

3) Line 110: “..western blots probed anti-uL18 revealed..”

probed with?

4) Line 124: “…but no band corresponding asterisked band in…”

??

5) Fig 1, labelling: Ribs or Rbs ?

6) Line 163: Why was the strain Pgal-e43 chosen for these experiments?

7) Line 164: “A band co-migrating with the ribosomal uL18 band ..”

was that inferred by some other detection method, as for example by mass spectrometry, or by the expected running behavior of uL18? Does the running behaviour of the detected band fit with the predicted size of uL18?

8) Line 166: “.. we confirmed that band that..”

that the band?

9) Line 174: “This was anticipated, since repression of the 60S r-protein uL5 is known to provoke a buildup of extra-ribosomal uL18..”

uL5 is together with uL18 part of the 5S RNP, in this regard I am not sure whether that specific example allows a straight forward prediction for what is to be expected upon expression shut down of an unrelated large subunit r-protein as eL43.

10) Line 177, Fig 2.: “Repression of several 40S r-protein genes causes … “

Several 40S r-protein genes? In Fig 2 is just an analysis shown for eS4.

11) Line 177, Fig 2.: “..causes accumulation of extra-ribosomal uL18/L5, but not extra-ribosomal uL5/L11 …”

How was the specificity analysed of the antibody used for detection of uL5?

12) Line 187: “(C) Northern blots of fractions from ..”

Western blots of ..?

13) Line 195: “.. was found outside in the ribosome peaks..”

outside of ?

14) Line 198: “.. parent whether it grows glucose and galactose medium..”

parent strain? grows in ?

15) Line 199: “..the results in Fig 2 shows that..”

show?

16) Lines 202 - 205: Why were these large and small subunit protein expression mutants chosen?

17) Line 260: “No uL18 band was seen at the top of the gradient 260 after cycloheximide inhibition of protein synthesis (Fig 4A(ii))..”

The western signals shown in Fig 4A (ii) look a bit blurry. Was a technical replicate of this experiment performed?

18) Line 296: “To investigate if extra-ribosomal uL18 also accumulates when rRNA synthesis th is inhibited by the TOR-targeting drug rapamycin”

th ? Rapamycine has also quite a drastic effect on the r-protein mRNA levels and thus on r-protein production. A conditional mutant of a gene coding for an RNA polymerase I subunit might have addressed the question asked more clearly.

19) Lines 310 and following: I was wondering if the authors also analysed the possible accumulation of uL18 bound 5S rRNA?

20) Line 332: “In the extremes, abolishment of eS4synthesis..”

eS4 synthesis?

21) Line 332 and following: “In the extremes, abolishment of eS4synthesis generates a response similar to that seen during repression of two 60S r-protein genes, while extra-ribosomal uL18 is borderline detectable during abrogation of eS31 synthesis (Fig 3). This gradient correlates with the abundance of 40S r-proteins in the 90S ribosomal particle, an early 40S assembly intermediate [19], suggesting that preventing early steps of pre-40S precursor assembly have the strongest effect on accumulation of extra-ribosomal uL18.”

Was the subunit imbalance phenotype compared between expression mutants of eS4 and eS31? The respective penetrance of the small subunit synthesis phenotype might well correlate with the observed effects on uL18. The authors suggest in the abstract and here in the discussion that specifically inhibition of early steps in small subunit assembly leads to the observed effects on uL18. That assumption would have been much strengthened if other 40S r-protein expression mutants with late 40S maturation phenotypes which were not detected in early 40S precursors would have been analysed (e.g. expression mutants of uS2, uS3, uS5 or others). I would also see the authors attempt to interprete the data from reference 19, Fig 7, in regard to the “abundance” of r-proteins in 90S pre-ribosomes as inappropriate, since the authors of reference 19 for probably good reasons use a cut-off to just deduce the presence of the respective proteins.

22) Line 348: “Since two 60S r-proteins (uL4 and uL24) bind to Domain 1 ..”

It is unclear to me why to list these two r-proteins here, but not other r-proteins which clearly also bind to LSU rRNA domain 1.

23) Line 367: “Since the major features of pathways for ribosomal assembly evolutionarily conservation, …”

?

24) Line 373: “The difference between the ratio of co-transcriptional and post-transcriptional is evident from ..”

Insert here “processing”?

25) Line 351 and following: “We therefore posit that the simplest explanation for the buildup of extra-ribosomal uL18 during inhibition of 40S assembly is that the folding of the 60S part of the early rRNA transcript is influenced by 40S r-proteins that bind to rRNA prior to separation of the subunit moieties of the emerging rRNA transcript.”

I am not sure whether this is really the simplest explanation. The effects described here were observed after rather long r-protein depletion times (6 -21 hours according to Materials and Methods). Effects on small subunit synthesis start to be detectable much earlier (after one to two hours in RNA pulse analyses) and after 6 hours and more depletion times quite a massive subunit imbalance can be established in small subunit r-protein expression mutants (see for examples the authors own publication in Life Science Alliance, 2019). That, together with the clear reduction in cellular amounts of functional ribosomes can have various consequences on the balanced production of each of the other ribosomal proteins, including the ones of the large subunit. Did the authors also test effects on extra-ribosomal uL18 accumulation after shorter incubation times in glucose medium at which effects on 40S production just start to be established (1-2 hours)? That might help to distinguish if the observed effects on uL18 are primary consequences of an altered early SSU maturation state or due to secondary effects on the translation status of the cell. Related to this point, see also point 21 in regard to the authors choice of a conditional eS31 expression mutant to control for effects on uL18 by a late SSU assembly defect.

6. PLOS authors have the option to publish the peer review history of their article (what does this mean?). If published, this will include your full peer review and any attached files.

Reviewer #1: Yes: John Woolford

Reviewer #2: No

---

## [Author Response · Author response to Decision Letter 0]

22 Dec 2019

Rebuttal for manuscript [PONE-D-19-24366]

We thank the reviewers for constructive criticisms. We have extensively revised the manuscript to accommodate the reviewer’s questions and suggestions. We hope that these changes will make the manuscript acceptable for PLOS one.

We agree that some of the additional experiments suggested might add interesting aspects to the manuscript. Unfortunately, logistic circumstances make further experiments impossible.

Reviewer #1: This manuscript by Rahman, Shamsuzzaman and Lindahl is an interesting follow-up to a previous paper from the Lindahl group showing interdependence of assembly of large and small ribosomal subunits in yeast. Here they demonstrate that ribosomal protein uL18, aka L5, accumulates outside of ribosomes when assembly of either large or small subunits is blocked by depleting a ribosomal protein from one or the other subunit. This was visualized by gradient sedimentation of L5 in extra-ribosomal fractions. Of interest is how varying effects of depletion of different r proteins reflects the order and mechanism of protein and subunit assembly, including effects resulting from post-transcriptional assembly. The conclusions are sound and the results are interesting. However, the story might be more impactful if the following were added:

(1 Include a figure showing in more detail which SSU and LSU r proteins are assembled early and which much later, and thus why depletion of some might have a greater effect on L5. This is explained in the text but would be clearer to nonaficionados if a figure/cartoon were added.

Good suggestion. We have added a model figure (Fig 8) and revised the text to coordinate it with the new figure.

 (2 L5 accumulates in fractions at the top of the gradient. Are 5S rRNA, L11, Rpf2 and Rrs1 also present in the same complex with L5? CO-IP from gradient fractions will demonstrate whether this is the case. This is potentially important since the authors report accumulation of extra-ribosomal L5 when Rpf2 or Rrs1 are depleted (Fig.4D).

As indicated above, additional experiments are not possible. However, as summarized in the revised manuscript, our existing evidence excludes the possibility that protection of extra-ribosomal uL18 requires the assembly of the complete uL18, uL5, Rfp2, Rrs1, and 5S rRNA “pre-docking” particle. 

Minor issues

(1 Some figures need to be better labeled and there are numerous places where a word or two need to be added or changed.

We have relabeled all figures according to the reviewer’s suggestion.

Line 56: Aren’t some r proteins not essential, i.e. not completely required for subunit assembly?

Correct, a subset of yeast r-proteins are not essential for growth under standard laboratory conditions and are thus not required for assembly of a (minimally?) functional ribosome. We have revised the text to make the distinction between essential and non-essential r-proteins.

Line 119 and elsewhere: a detail perhaps: is the symbol in the figures a star or an asterisk?

Correct, the symbols at the non-uL18 band in the figures is a star. The text has been corrected to reflect this.

Line 151 and elsewhere: the term GAL1/10 promoter should be capitalized and italicized.

Thanks; corrected.

Line187: what is meant by Northern blots of fractions were probed? Don’t you mean just “blots”?

We mean “western blots”. The error was corrected.

Line 367: is ther a word or two missing?

Thanks. Text corrected.

Figures: Does more than one lane of blot data correspond to consecutive gradient fractions, e.g. lanes 3,4 and 5,6 in Fig.1? If so, please explain in the figure legends.

So indicated.

Figure 2B: the text states that L11 was assayed, but it is not shown.

uL5/L11 panel was added to Fig 2B (now Fig 2C).

(2 In the Introduction, lines 64-66, the authors need to correct their description of the experiment done by Dehmukh et al. They got trapped by the nomenclature monster! R protein L16 was depleted, which is NOT the same as uL5, as stated. This is important because uL5 aka L11 is in an assembly subcomplex with uL18 aka L5, whereas L16 is not.

We have retraced our way through the “nomenclature monster” and still end up with the L16 (anno 1993; i.e. Deshmukh et al) being identical to uL5 (anno 2014): Table 1 of the 1997 paper by Mager et al equates old L16 with L11A (Mager WH, Planta RJ, Ballesta JG, Lee JC, Mizuta K, Suzuki K, et al. A new nomenclature for the cytoplasmic ribosomal proteins of Saccharomyces cerevisiae. Nucleic Acids Res. 1997;25(24):4872-5). Table 2 of Ban et al further equates L11 with uL5 (http://dx.doi.org/10.1016/j.sbi.2014.01.002 ). Our procedure is validated by converting L1 (anno 1993) > uL18 (anno 2014): Mager et at Table 1 converts L1 (anno 2993) to L5 (anno 1997) which is further converted by Table 2 of Ban et al to uL18 (anno 2014). Furthermore, the curators of SGD agree with our conversion of L16 to L11/uL5 since their entries for RPL11 A and B (1997 nomenclature) include a paper on disruption of ribosome formation during L16 depletion from the Woolford lab (https://www.ncbi.nlm.nih.gov/pubmed/3282992). We are thus convinced that our nomenclature conversion is correct. We also note that uL5/L11 does not accumulate extra-ribosomally in proportion with uL18/L5 (figs 2B, S2). Thus, uL5/L11 is not an obligatory component of the extra-ribosomal complex protecting uL18/L5 against rapid degradation. See also our response to the reviewer’s suggestion for an IP experiment.

Reviewer #2: Rahman et al. report in the manuscript PONE-D-19-24366 on the results of a series of experiments which provide evidence for the extra-ribosomal accumulation of the large ribosomal subunit protein uL18 upon inhibition of small ribosomal subunit assembly in S. cerevisiae. Their conclusions are based on quantitative interpretation of western blotting analysis after sucrose gradient fractionation of cellular extracts prepared from yeast ribosomal protein expression mutants. In this regard the exact approach chosen might be better described.

We have expanded Materials and Methods to clarify our strategy and experiment execution.

The authors refer to reference 9, in which an indirect immuno-detection is described using an enzymatic reaction for secondary antibody visualisation. These detection approaches tend to produce saturation artefacts and it might be therefor important to know whether titration of a reference sample was used in the individual western blots to determine the dynamic range of the detected signal. 

The antisera were titrated against increasing amounts of a standard lysate in Figure S6 in Gregory et al (Life Sci Alliance. 2019;2(2). Epub 2019/03/07. doi: 10.26508/lsa.201800150. PubMed PMID: 30837296; PubMed Central PMCID: PMCPMC6402506). For the reviewer’s convenience the figure is reproduced at the end of our rebuttal. Based on these curves, the amount of extract loaded on the sucrose gradient, the number of sucrose gradient fractions into which the r-proteins are distributed, and the volume from each fraction, we calculated that our experiments are within the dynamic range of the antibodies. 

For the sucrose gradient fractionation, in several experiments all fractions were analysed. In these cases the ribosomal proteins could be detected in four to six of the “Top” fractions and in up to eleven of the faster sedimenting “Ribosomes” fractions. In several other analyses only one or two of the “Top” fractions and either two or three of the “Ribosomes” fractions of the sucrose gradients were analysed to deduce the ratio of extra-ribosomal to ribosomal r-proteins as uL18. How was it ensured that these fractions contain all of the extra-ribosomal and ribosomal pools of uL18, or are representative for them?

Quantification of extra-ribosomal uL18 based on the limited number of fractions was justified by the following analysis of blots loaded with all sucrose gradient fractions. We calculated extra-ribosomal uL18/total uL18 using (i) all fractions and (ii) two fractions from the top and three fractions from the 60S-80S peaks. The results (Table S2 - new) show that there is semi-quantitative agreement between the two methods. We conclude that our approach is adequate for comparing the level of extramural uL18 after repressing different r-protein genes. When comparing results determined in different ways (e.g. quantifying one versus two top fractions), we have normalized the results to a reference sample.

The western blot results shown in Fig 3A , 3C(iii) and 4D look quite cropped. That complicates interpretation of the data, especially since the uL18/L5 antibody used in this study cross-reacts with a protein running just a bit slower than uL18/L5. The identity and running behavior of the band detected in the “Top” fractions in 3A for the parental strain is, for example, in the

actual representation for me ambiguous. Thus, less cropped figures might help the reader in these cases.

We have re-cropped the images in Fig 3A showing larger sections. Furthermore, uncropped images for all figures are now shown in Fig S1-S3.

For several of the experiments (Fig 3A +B) I could not find the exact incubation times in glucose containing medium which were applied to repress the expression of individual r-proteins. In the materials and methods section it is stated that cultures were shifted for 6 to 21 hours to glucose containing medium. In one experiment (Fig. 2) the control strain is shifted to glucose containing medium for a different time period (16 hours) than the two r-protein expression mutants analysed (6 hours and 8 hours). Why was that done, and can we be sure that this might not affect the outcome of the experiments? 

The determination of extra-ribosomal uL18 relative to total uL18 is complicated by the fact that “old ribosomes” (made before the shift to glucose) are stable. Moreover, formation of 60S subunits continue after repression of 40S genes, but at changing rates as the growth rate of the culture decreases after the media shift. We think that the best approach to estimate the extra-ribosomal uL18 pool size therefore is to sample the culture after it has gone through at least two doublings after the shift, at which time the fraction of uL18 in extra-ribosomal fractions is in equilibrium (see new Fig 4). Since the growth rate is different after the shift the cultures were thus incubated for different times in glucose medium. The actual times are now shown in Fig 3A. Regarding the parent strain, the time in glucose is not important, since the density of the culture was kept in the exponential growth range by periodical dilution with prewarmed medium.

Arguably, the net rate of formation of extra-ribosomal uL18 could have been determined more accurately using radioactive pulse labeling followed by immune-precipitation from total extracts and the extra-ribosomal fractions (top of the gradient). However, we did not expect to learn more from this more complicated approach than we did my letting the uL18 population come to equilibrium. 

At least from the data shown in Fig 3C for Pgal-eS31 I got the impression that the level of free uL18/L5 decreases with prolonged incubation times in glucose containing medium.

Only occasionally did we see uL18 bands at the top of the gel in Pgal-eS31. As indicated, in Fig 3B, extra-ribosomal uL18 is not significantly different from the value in the parent. In the original submission Fig 3B we indicated that eS31 was different from the parent. This was an error for which we apologize. All calculations have been reexamined to assure accuracy of the revised Fig 3B. We have also included a matrix of significance, by t-test, between the level of extra-ribosomal uL18 in the different strains.

Recently, changes in levels of proteins in several large and small subunit r-protein expression mutants were measured by mass spectrometry proteome wide (https://doi.org/10.1016/j.molcel.2018.10.032). Did the authors analyse these published datasets for a possible excess of uL18/L5 over other large subunits proteins as ul4/L4 or uL5/L11? That might give further support to the authors conclusions by an analysis performed with an independent experimental approach.

Unfortunately, the paper by Cheng et al does not have data on uL18 and our paper (Gregory et al) does not analyze the same protein set as used in the current manuscript. 

A few other comments and remarks:

1) Line 52: “..also requires in excess of 250 ribosomal assembly factors” Does that refer to the stochiometry of components?

We are referring to the number of factors. To avoid confusion, we replaced “In excess” with ”more than”.

2) Line 54: “..while others are specific to the formation of one the ribosomal subunits” of one of ?

Thank you. Thee text was revised.

3) Line 110: “..western blots probed anti-uL18 revealed..” probed with?

Correct. Thank you

4) Line 124: “…but no band corresponding asterisked band in…” ??

Corresponding to ---Thank you

5) Fig 1, labelling: Ribs or Rbs ?

The figure was relabeled to accommodate comments from reviewer 1.

6) Line 163: Why was the strain Pgal-e43 chosen for these experiments?

We have previously used Pgal-eL43 in a number of experiments (Gregory et al; Shamsuzzaman M et al. PLoS One 12: e0186494. doi:10.1371/journal.pone.0186494; Thapa et al Mol Biol Cell 24: 3620–3633. doi:10.1091/mbc.e13-02-0097). The choice of Pgal-eL43 was originally somewhat arbitrary, but it has become one of several standard strains often used in our lab.

7) Line 164: “A band co-migrating with the ribosomal uL18 band ..” was that inferred by some other detection method, as for example by mass spectrometry, or by the expected running behavior of uL18? Does the running behaviour of the detected band fit with the predicted size of uL18?

We are referring to co-migration during gel electrophoresis, which has now been specified. The band migrates in the SDS gel as expected from its molecular weight. More importantly the identity of the band was confirmed in Materials and Methods (Fig 1 and accompanying text).

8) Line 166: “.. we confirmed that band that..” that the band?

Correct. Thank you

9) Line 174: “This was anticipated, since repression of the 60S r-protein uL5 is known to provoke a buildup of extra-ribosomal uL18..” 

uL5 is together with uL18 part of the 5S RNP, in this regard I am not sure whether that specific example allows a straight forward prediction for what is to be expected upon expression shut down of an unrelated large subunit r-protein as eL43.

The result was expected, because cessation of either protein virtually abolishes assembly of new 60S subunits.

10) Line 177, Fig 2.: “Repression of several 40S r-protein genes causes … “

Several 40S r-protein genes? In Fig 2 is just an analysis shown for eS4.

Thank you. We already noticed this misstatement and corrected it

11) Line 177, Fig 2.: “..causes accumulation of extra-ribosomal uL18/L5, but not extra-ribosomal uL5/L11 …”

How was the specificity analysed of the antibody used for detection of uL5?

First, the N-terminal sequence is unique amount r-proteins. Second, the electrophoretic mobility is as expected from the size, and third the protein reacting with the antiserum behaves like a 60S protein in terms of co-sedimenting with the 60S subunit and declining when 60S assembly is inhibited by repression of genes for uL4, uL18, and eL40 (Gregory et al).

12) Line 187: “(C) Northern blots of fractions from ..”

Western blots of ..?

Correct. Thank you.

13) Line 195: “.. was found outside in the ribosome peaks..”

outside of ?

Correct. Thank you.

14) Line 198: “.. parent whether it grows glucose and galactose medium..”

parent strain? grows in ?

Correct. Thank you.

15) Line 199: “..the results in Fig 2 shows that..”

show?

Correct. Thank you.

16) Lines 202 - 205: Why were these large and small subunit protein expression mutants chosen?

 eS6, uS17, eS19

 eS6, uS17, and eS19 were picked because mutation in the human orthologues have disease consequences. The other two strains, eS4 and eS31, were picked because they are incorporated early and late, respectively, during assembly of the 40S subunit. This is now explained in the text.

17) Line 260: “No uL18 band was seen at the top of the gradient 260 after cycloheximide inhibition of protein synthesis (Fig

4A(ii))..”

The western signals shown in Fig 4A (ii) look a bit blurry. Was a technical replicate of this experiment performed?

We did not do a direct technical replicate, but the results agreed with other experiments.

18) Line 296: “To investigate if extra-ribosomal uL18 also accumulates when rRNA synthesis th is inhibited by the TORtargeting

drug rapamycin”

th ? Rapamycine has also quite a drastic effect on the r-protein mRNA levels and thus on r-protein production. A conditional

mutant of a gene coding for an RNA polymerase I subunit might have addressed the question asked more clearly.

Conditional RNA pol I mutants are fairly sick and take a long time to express their phenotype, so we do not think that experiments with RNA Pol I mutants would be useful. We have revised the text to include the results (degradation of cytoplasmic ribosomes) of Pestov et al (see revised reference list).

19) Lines 310 and following: I was wondering if the authors also analysed the possible accumulation of uL18 bound 5S rRNA?

We did not. We considered it, but concluded that we would not add to the observations of Deshmukh et al.

20) Line 332: “In the extremes, abolishment of eS4synthesis..”

eS4 synthesis?

Yes, thank you. We already corrected this typo.

21) Line 332 and following: “In the extremes, abolishment of eS4synthesis generates a response similar to that seen during

repression of two 60S r-protein genes, while extra-ribosomal uL18 is borderline detectable during abrogation of eS31 synthesis

(Fig 3). This gradient correlates with the abundance of 40S r-proteins in the 90S ribosomal particle, an early 40S assembly intermediate [19], suggesting that preventing early steps of pre-40S precursor assembly have the strongest effect on accumulation of extra-ribosomal uL18.”

Was the subunit imbalance phenotype compared between expression mutants of eS4 and eS31? 

We have added a statistical analysis comparing the different mutants (Fig 3C)

The respective penetrance of the small subunit synthesis phenotype might well correlate with the observed effects on uL18. The authors suggest in the abstract and here in the discussion that specifically inhibition of early steps in small subunit assembly leads to the observed effects on uL18. That assumption would have been much strengthened if other 40S r-protein expression mutants with late 40S maturation phenotypes which were not detected in early 40S precursors would have been analysed (e.g. expression mutants of uS2, uS3, uS5 or others). I would also see the authors attempt to interprete the data from reference 19, Fig 7, in regard to the “abundance” of r-proteins in 90S pre-ribosomes as inappropriate, since the authors of reference 19 for probably good reasons use a cut-off to just deduce the presence of the respective proteins.

Nevertheless eS4, eS6, uS11, and eS19, but not eS31, are all above the cut-off and therefore must be concluded to be among the most abundant 40S r-proteins in the 90S.

We agree that analysis of repression of the synthesis of more proteins could have strengthened, or better tested, our model. However, we do think our manuscript is important, because it uncovers a previously unknown interaction between 40S assembly and the kinetics of 60S assembly. A deeper investigation of the many interesting questions will be complex and is beyond the scope of this manuscript. 

22) Line 348: “Since two 60S r-proteins (uL4 and uL24) bind to Domain 1 ..”

It is unclear to me why to list these two r-proteins here, but not other r-proteins which clearly also bind to LSU rRNA domain s1.

We eliminated this from the text.

23) Line 367: “Since the major features of pathways for ribosomal assembly evolutionarily conservation, …” ?

We revised the text

24) Line 373: “The difference between the ratio of co-transcriptional and post-transcriptional is evident from ..”

Insert here “processing”?

No, here we actually meant what we wrote.

25) Line 351 and following: “We therefore posit that the simplest explanation for the buildup of extra-ribosomal uL18 during inhibition of 40S assembly is that the folding of the 60S part of the early rRNA transcript is influenced by 40S r-proteins that bind to rRNA prior to separation of the subunit moieties of the emerging rRNA transcript.”

I am not sure whether this is really the simplest explanation. The effects described here were observed after rather long rprotein depletion times (6 -21 hours according to Materials and Methods). Effects on small subunit synthesis start to be detectable much earlier (after one to two hours in RNA pulse analyses) and after 6 hours and more depletion times quite a massive subunit imbalance can be established in small subunit r-protein expression mutants (see for examples the authors own publication in Life Science Alliance, 2019). That, together with the clear reduction in cellular amounts of functional ribosomes can have various consequences on the balanced production of each of the other ribosomal proteins, including the ones of the large subunit. Did the authors also test effects on extra-ribosomal uL18 accumulation after shorter incubation times in glucose medium at which effects on 40S production just start to be established (1-2 hours)? That might help to distinguish if the observed effects on uL18 are primary consequences of an altered early SSU maturation state or due to secondary effects on the translation status of the cell. Related to this point, see also point 21 in regard to the authors choice of a conditional eS31 expression mutant to control for effects on uL18 by a late SSU assembly defect.

This of course depends on how “simple” is interpreted. We think that it is almost inescapable that our model must contribute the changes in 60S assembly kinetics and the accumulation of extra-ribosomal uL18, but have inserted comments as to potential effects of other parameters.

---

## [Editor Report · Decision Letter 1]

8 Jan 2020

Interaction between the assembly of the ribosomal subunits: Disruption of 40S ribosomal assembly causes accumulation of extra-ribosomal 60S ribosomal protein uL18/L5

PONE-D-19-24366R1

Dear Lasse,

I hope you had a fabulous Christmas and a fantastic start of 2020. Apologies for the delay in processing the manuscript but I had made a promise to my wife that I would not be doing any work during the Christmas period and therefore only started reading the revised version a few days ago. I have have now had a chance to properly read the revised manuscript and I am happy to recommend the paper for publication.

With kind regards,

Sander Granneman

Academic Editor

PLOS ONE
---

## [Editor Report · Acceptance letter]

17 Jan 2020

PONE-D-19-24366R1 

Interaction between the assembly of the ribosomal subunits: Disruption of 40S ribosomal assembly causes accumulation of extra-ribosomal 60S ribosomal protein uL18/L5 

Dear Dr. Lindahl:

I am pleased to inform you that your manuscript has been deemed suitable for publication in PLOS ONE. Congratulations! Your manuscript is now with our production department. 

With kind regards,

on behalf of

Dr. Sander Granneman 

Academic Editor

PLOS ONE